# Performance Analysis of Anaerobic Digestion Coupled with Simultaneous Nitrification and Denitrification Process for Treating Alcohol Precipitation Wastewater of Chinese Patent Medicine

**Xiaofeng Jia [1], Hanxin Fan [2], Jiawei Liang [3,4], Jihua Dai [4,*], Yu Sun [1] and Wenning Mai [2]**

[1] School of Civil Engineering and Architecture, Zhengzhou University of Aeronautics, Zhengzhou 450015, China
[2] School of Ecology and Environment, Zhengzhou University, Zhengzhou 450000, China
[3] College of Public Health, Zhengzhou University, Zhengzhou 450001, China
[4] Research Center for Eco-Friendly Wastewater Purifying Engineering Technology of Henan Province, Henan Junhe Environmental Protection Technology Co., Ltd., Zhengzhou 450001, China
* Correspondence: daijihua2023@163.com

**Abstract:** The alcohol precipitation wastewater discharged from the production of Chinese patent medicine (CPM) has an extremely high chemical oxygen demand (COD) and poor biodegradability. In this study, the biological treatment method of anaerobic digestion coupled with simultaneous nitrification and denitrification (SND) was adopted to investigate its efficiency and to explore the mechanism of pollutant degradation in this process. The results showed that after 220 days of debugging, the coupled process operated stably. The influent COD, total nitrogen (TN), ammonium ($NH_4^+$-N), and lignin concentrations were 21,000 mg/L, 400 mg/L, 200 mg/L, and 1800 mg/L, respectively. The removal efficiencies of COD, TN, $NH_4^+$-N, and lignin were 97%, 85%, 96%, and 75%, respectively. Spectral detection technology analysis revealed that the wastewater contained alkanes, olefins, phenols, alcohols, unsaturated organics, aromatic compounds, and humic acids. After the treatment by each unit of the process, the three-dimensional fluorescence intensity decreased by 86%; the standard volume of fluorescence area integration declined by 78%; the stretching vibration band of aromatic compounds showed peak splitting; and the molecular weight parameter value in the ultraviolet region increased. These findings demonstrated that the humic acid substances in the wastewater were degraded, and the effect of removal of the macromolecular organic matter was remarkable.

**Keywords:** CPM wastewater; alcohol precipitation; anaerobic digestion; SND; operational performance

## 1. Introduction

According to the National Bureau of Statistics of China, the yield of Chinese patent medicines in China exceeded 2.3 million tons in 2021 alone. However, the production process of Chinese patent medicines may discharge a large amount of wastewater [1], which has a high concentration of chemical oxygen demand (COD), large chroma, and high biological toxicity. The direct discharge of this untreated wastewater into the water bodies would result in severe pollution in the aquatic environment [2]. During the production process of Chinese patent medicines, the main sources of wastewater are the water used in cleaning equipment and machines, workshop scraps and waste, workshop floors and auxiliary sections, as well as the extraction workshop and domestic wastewater [3,4]. Among these sources, the alcohol precipitation wastewater released from the extraction workshop has the highest COD and pollution load, which is the most serious concern in the field of Chinese patent medicine wastewater treatment.

Currently, many technologies are being applied for the treatment of this wastewater. However, the applied technique and performance vary greatly for different kinds of wastewater [1,5]. Overall, the treatment processes include biological methods, physicochemical methods, and combined physicochemical-biological methods. The physicochemical methods have been reported to show a good removal rate of suspended solid (SS) and chroma in wastewater, while the removal of insoluble COD was poor [6–10]. However, the cost of implementing these methods is high. For example, Liu et al. [11] combined iron–carbon micro-electrolysis with Fenton oxidation and coagulation precipitation for the treating concentrated wastewater of the Chinese patent medicine industry. The treatment reduced the COD of the wastewater from 24,600 to 2726 mg/L. However, the cost of this treatment was above CNY 7.67 per ton of water (excluding labor costs). On the other hand, biological treatment showed a high removal rate of COD and SS for Chinese patent medicine wastewater, with better biodegradability [12–15]. The expanded granular sludge bed (EGSB) reactor has been reported to be suitable for the treatment of traditional Chinese medicine wastewater, even with high organic loading rates (OLRs) of up to 13 kg COD/(m$^3$·d), and the COD removal was good enough (>90%) for practical application [16]. Similarly, a novel double circle (DC) anaerobic reactor derived from an IC reactor (adding external circulation) also achieved efficient treatment of traditional Chinese medicine wastewater, with a high COD removal of 96.87% and for OLRs ranging from 13 to 15 kg COD/(m$^3$·d) [17]. More recently, a pilot scale, novel, spiral symmetry stream anaerobic bioreactor (SSSAB) was applied for treatment of traditional Chinese pharmaceutical wastewater, and was compared with the IC reactor working in the same conditions [18]. Compared to the IC reactor, SSSAB was able to handle much higher OLRs and achieved higher COD removal efficiency and volumetric removal rate (VRR) [18]. However, the biodegradability of Chinese patent medicine wastewater is generally low, and has a strong impact on simple biochemical treatment, resulting in an unstable biological treatment effect. Therefore, in recent years, various combined physicochemical–biological or advanced biological methods were developed for more effective and efficient wastewater treatment [19–23]. However, the concentration and biodegradability of the wastewater determined the complexity of the treatment process. Higher concentration led to a longer treatment process, and less biodegradable wastewater led to a complex treatment process.

To sum up, the alcohol precipitation wastewater is one of the main sources of high concentration and low biodegradable pharmaceutical wastewater. There are many reports regarding the effects of alcohol sedimentation wastewater [24–26]. However, no research has been reported for the treatment of alcohol precipitation wastewater. Therefore, the present study was undertaken to investigate the characteristics of the alcohol precipitation wastewater and to develop a new treatment process for reducing its toxicity and improving biodegradability. The findings of this study would open a new direction for the treatment of high-concentration Chinese patent medicine wastewater.

This study proposes the coupling of anaerobic digestion with the SND process for the biochemical treatment of alcohol precipitation wastewater. Anaerobic digestion is an ideal and widely used technology for treating high-concentration organic wastewater [27], which not only saves energy but also produces energy. On the other hand, SND technology uses nitrifying and denitrifying microorganisms to simultaneously perform the nitrification and denitrification processes in the same environment, thereby providing the advantages of low energy consumption and carbon source saving [28]. SND has been reported to be a promising method for COD and nitrogen removal from wastewater [29,30]. Compared to the traditional biological treatment processes, SND requires less aeration for operation and low dosages of acids and alkalis.

Alcohol precipitation wastewater has not only a high COD concentration, but also high $NH_4^+$-N concentration. Therefore, anaerobic digestion and SND processes were combined in this study to take advantages of both processes. The key operating parameters of the process were obtained through long-term debugging and operation. Subsequently, the changes in the main components of the wastewater were observed through spectral

detection technology. In addition, the mechanism of pollutant degradation in this process was ascertained.

## 2. Materials and Methods

### 2.1. Experimenalt Device

Figure 1 shows the experimental device. Column 1# shown in the figure was an anaerobic digestion unit. Column 2# and Column 3# were SND units. All the three columns were up-flow reactors made of plexiglass, with a stirrer inside each column. Each reactor was mostly cylindrical. The size of the bottom reaction zone was $\Phi100$ mm Í600 mm, and the size of the top sedimentation zone was $\Phi200$ mm Í200 mm. The built-in three-phase separator in the sedimentation zone was capable of separating sludge, water, and gas, without any sludge loss. Each reactor was wrapped with a temperature-controlled heating tape and insulation cotton to maintain the temperature required for operation. Moreover, each reactor had a built-in portable pH meter to monitor the change in pH during operation. The effluent of Column 1# as well as effluent and recirculation water of Column 3# were introduced into Column 2# through the bottom zone of Column 2# (Figure 1). After overflowing from the upper sedimentation zone, the effluent of Column 2# was introduced to Column 3# through the bottom zone of Column 3#. The temperature and pH of Column 1# were maintained at 33–35 °C and 6.8–7.2, respectively. The temperature of both Column 2# and Column 3# was kept in the range of 23~25 °C, with the pH ranging between 7.3 and 7.8. The bottom aeration which kept the DO below 2 mg/L was set in Column 3#, and the change in DO concentration during operation was monitored by a portable DO meter.

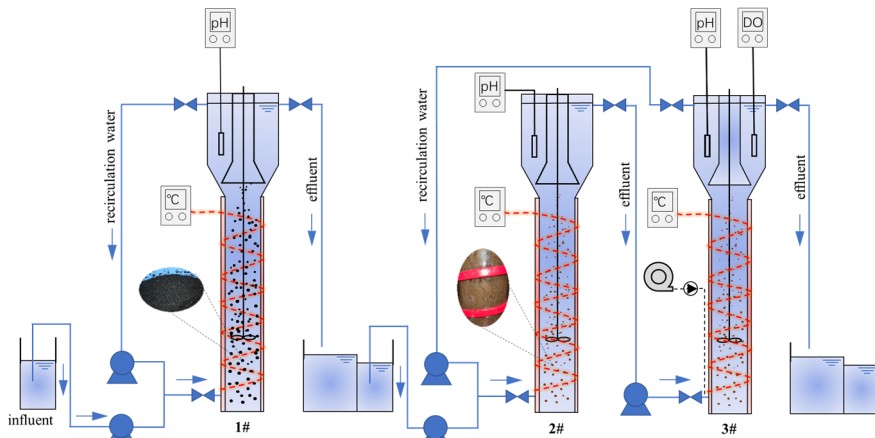

**Figure 1.** Schematic diagram of the experimental device: 1#: anaerobic digestion unit; 2# and 3#: SND unit.

### 2.2. Materials

#### 2.2.1. Wastewater

The wastewater used in this study was taken from a Chinese patent medicine manufacturing enterprise in Shandong Province, China. Three batches of wastewater samples were collected from the enterprise. Table 1 shows the water quality index of the alcohol precipitation wastewater. The enterprise mainly produced Chinese patent medicine products such as Ganmaoling Granules and Banlangen Granules, with botanical materials such as Sanchaku and Banlangen. During the experiment, distilled water was used to dilute the alcohol precipitation wastewater according to the increasing concentration of the influent water, and the proportion of the alcohol precipitation wastewater was gradually increased.

**Table 1.** The water quality index of the alcohol precipitation wastewater.

| Indexes | Values | Indexes | Values |
|---|---|---|---|
| COD (mg/L) | 250,000 ± 17,552 | Total salinity (mg/L) | 22,860 ± 1830 |
| TN (mg/L) | 3196 ± 209 | SS (mg/L) | 70,570 ± 4290 |
| $NH_4^+$-N (mg/L) | 2476 ± 337 | pH | 4.5~5.5 |

Note: SS: Suspended solid.

### 2.2.2. Seed Sludge

The seed sludge used in the anaerobic digestion unit was the granular sludge collected from the anaerobic reactor of a starch factory in Kaifeng, China. The amount of mixed liquor suspended solids (MLSS) in the seed sludge was 150 g/L, and the ratio of mixed liquor volatile suspended solids (MLVSS) to MLSS was 0.75. The amount of inoculum accounted for 30% of the total volume of the reactor.

The inoculum sludge used in the SND unit was the aerobic activated sludge collected from the secondary sedimentation tank of a municipal sewage treatment plant in Zhengzhou, China. The amount of MLSS of the inoculum sludge was 10 g/L, and the MLVSS/MLSS was 0.72. Before inoculation, the inoculum sludge was aerated only with clean water for 24 h to activate the sludge performance, thereby facilitating the cultivation and acclimation. After inoculation, the MLSS in columns 2# and 3# was 5000 mg/L.

### 2.3. Debugging Process

This study was divided into two stages. The first stage was the independent debugging process of each unit. The COD removal in the anaerobic digestion unit was explored by adjusting the inflow and COD concentration of the influent. The optimal operating parameters for the SND units were determined by adjusting the influent inflow, influent TN concentration, reflux ratio of Column 2# (i.e., the ratio of Column 3# reflux water to Column 2# influent), and the DO concentration of Column 3#.

In the second stage, the anaerobic digestion and the SND process were coupled. The overall operational performance and pollutant removal efficiency of the anaerobic digestion coupled with the SND process were investigated for treating the alcohol precipitation wastewater of Chinese patent medicine.

### 2.4. Analytical Methods

The analytical methods used for the characterization are summarized in Table 2.

**Table 2.** Detection and analysis methods.

| No. | Indexes | Methods | Standards |
|---|---|---|---|
| 1 | COD | Potassium chromate method | GB 11914-1989 |
| 2 | TN | Alkaline potassium persulfate digestion ultraviolet spectrophotometry | HJ 636-2012 |
| 3 | $NH_4^+$-N | Nessler's reagent spectrophotometry | HJ 535-2009 |
| 4 | $NO_3^-$-N | UV spectrophotometry | HJ/T 346-2007 |
| 5 | DO | Electrochemical probe method | HJ 506-2009 |
| 6 | VFA and ALK | VFA/ALK combined titration method | Q/YZJ10-03-02-2000 |
| 7 | MLSS and MLVSS | Weighing method | GB 11901-89 |
| 8 | pH | Glass electrode method | GB/T 6920-1986 |
| 9 | Lignin | Spectrophotometry | DB 22/T 1670-2012 |
| 10 | Acute toxicity | Luminescent bacteria method | GB/T 15441-1995 |

Note: VFA: Volatile fatty acid; ALK: Alkalinity.

Dissolved organic matter (DOM) was quantitatively analyzed by three-dimensional excitation–emission matrix fluorescence spectroscopy (3D-EEM) [31]. The chemical characterization of the wastewater was carried out by Fourier-transform infrared spectroscopy (FTIR) [32]. The FTIR was recorded by a Nicolet iS20 spectrometer (Thermo Scientific, Waltham, MA, USA) at wave numbers ranging from 400 to 4000 cm$^{-1}$. For the FTIR analysis, KBr pellets were prepared by mixing the wastewater samples with KBr. In addition, pollutant removal in each unit was detected by ultraviolet visible (UV–Vis) absorption spectroscopy [33]. The absorbance was measured with a TU-1900 spectrophotometer (PERSEE, China), in the range of 200–800 nm.

## 3. Results and Discussion

### 3.1. COD Removal Efficiency during Anaerobic Digestion

The debugging process of the anaerobic digestion unit was divided into three stages. Figure 2 shows the changes in influent and effluent COD, the COD removal rate, and the COD loading rate in each stage.

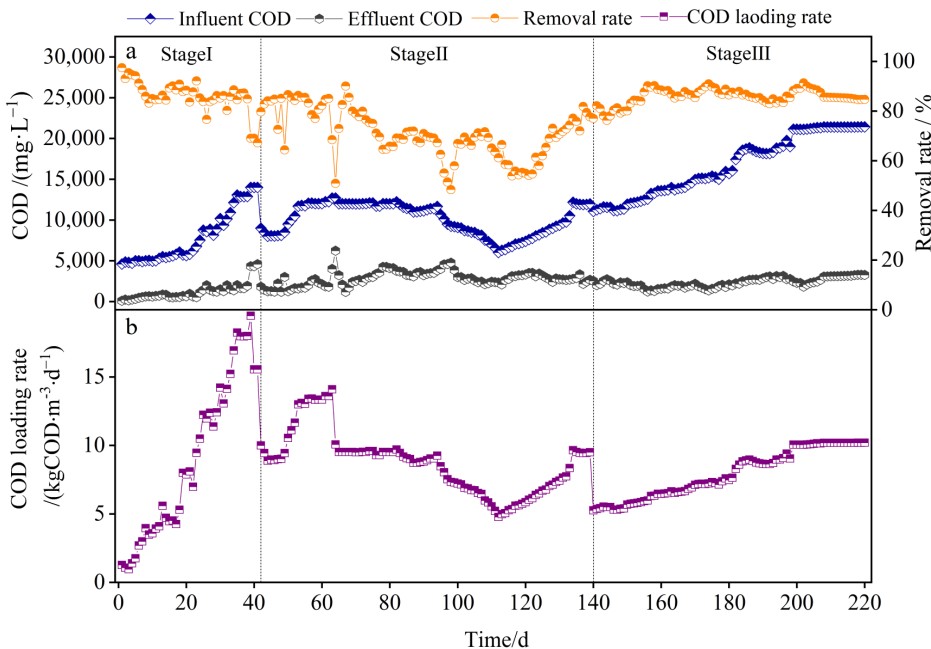

**Figure 2.** COD removal in the anaerobic digestion unit during the debugging process: (**a**) influent and effluent COD, the COD removal rate (**b**) the COD loading rate..

Stage I was from day 1 to day 41. During stage I, the influent inflow and influent concentration were alternately increased to examine the operating performance of the reactor. The COD of the raw water was diluted to 5000 mg/L. The volume load of 1.0 kg COD/(m$^3$·d) was used for the start-up of the operation. As shown in Figure 2, the COD removal rate of the reactor in the first 6 days remained above 90%, due to the dilution of clear water in the reactor. By the 15th day, the COD removal rate remained above 85%. Afterwards, the loading was increased gradually. Each increase in loading experienced 2~4 influent cycles. During the increase in loading, the reactor was observed to be affected by the shock of the pollutant concentration. Consequently, the effluent COD concentration first increased and then started to decrease. In addition, floating sludge appeared in the sedimentation area of the reactor, and granular sludge was found in the effluent and circulating water. The presence of granular sludge may be attributed to several reasons. During the start-up process, granular sludge unsuitable for the wastewater quality may have been eliminated. Moreover, the low substrate concentration may have led to the death of microorganisms inside the granular sludge, due to the lack of substrates for growth. This granular sludge may have flowed out with the effluent after cavitation. On the 39th day,

when the influent COD concentration and volume loading were increased to 14,350 mg/L and 19.5 kg COD/(m³·d), respectively, the effluent COD suddenly reached 4333 mg/L. Furthermore, the COD removal rate dropped sharply to 69.8%, resulting in the acidification of the reactor, with the effluent VFA reaching 20 mmol/L. These changes may be attributed to the abrupt increase in the influent COD concentration. To avoid reactor collapse due to continuous acidification, sodium hydroxide (1 mol/L) was added into the reactor until its pH was 7.2, reducing influent inflow and influent concentration, and increasing effluent reflux [34].

Stage II spanned from day 42 to day 139. During stage II, the tolerance of the reactor for wastewater was investigated by maintaining the influent concentration and increasing the influent inflow. From the 42nd to the 52nd day, the influent COD of 9000 mg/L was maintained for acidification recovery, which resulted in the increase in the COD removal rate, to 85%. The influent inflow increased from 5 L/d to 7 L/d between the 52nd and 90th day, while the influent COD was between 9000 and 11,000 mg/L. The COD removal rate dropped sharply to 50.9% and 48.3% on the 64th day and 98th day, respectively. The decrease may have happened due to the acclimatization process in the reactor, which led to the fluctuations in water quality, causing an inhibitory effect on the microorganisms in anaerobic sludge. After the 98th day, the COD removal rate was recovered through water replacement and reduction in the influent COD concentration. After the 118th day, the COD removal rate reached 75%. During stage III on the 140th day, the influent inflow in the reactor was 3 L/d. The debugging process was conducted with the influent COD concentration of 11,000 mg/L and the volume loading of 5 kg COD/(m³·d). The influent COD concentration was gradually increased, with a 0.5 kg COD/(m³·d) rise in volume loading each time. With the increase in volume loading, the COD removal rate decreased slightly. However, the COD removal rate started to increase gradually after the reactor adapted to the increase in volume loading. After 80 days of gradual loading increase, the influent COD reached 21,000 mg/L. At this time, the effluent COD was around 3200 mg/L, the COD removal rate was around 85%, and the volume loading was stable at around 10 kg COD/(m³·d). Afterwards, the anaerobic digestion process ran stably under these operating conditions.

The debugging process indicated that it was feasible to adopt anaerobic digestion for high-concentration alcohol precipitation wastewater. Moreover, more than 85% of the organic pollutants were removed, thereby reducing the subsequent biological treatment loading.

*3.2. COD Removal Efficiency during SND*

The debugging of the SND units was also divided into three stages. Figure 3 shows the changes in the COD of the influent and effluent, COD removal rate, and COD loading rate in each stage.

Day 1 to day 35 was the sludge acclimation stage. During this stage, the sludge was acclimatized by increasing the influent inflow and influent COD concentration. The influent COD was gradually increased from 500 mg/L to 3000 mg/L. After the 23rd day, the COD removal rate stabilized, in the range of 60–85% (Figure 3b). The proportion of the COD removal rate of Column 2# in the total removal rate gradually increased, from 40~50% to 65~75% in the early acclimation stage. This increase in the COD removal rate can be attributed to the activity of the denitrifying bacteria, which may have used most of the organic matter to proliferate and to carry out denitrification during acclimatization. The COD removal in Column 3# first decreased and then increased to 15~20%. This may be because the two columns were inoculated with the same activated sludge. The microbial community structures in the two columns were also similar during the early acclimatization stage. The organic carbon source consuming the heterotrophic bacteria in Column 3# utilized basically the same amount of COD as the microbes in Column 2#. However, the activity of the nitrifying bacteria consuming the inorganic carbon source increased gradually in the 3# column, leading to the decrease in the COD removal rate.

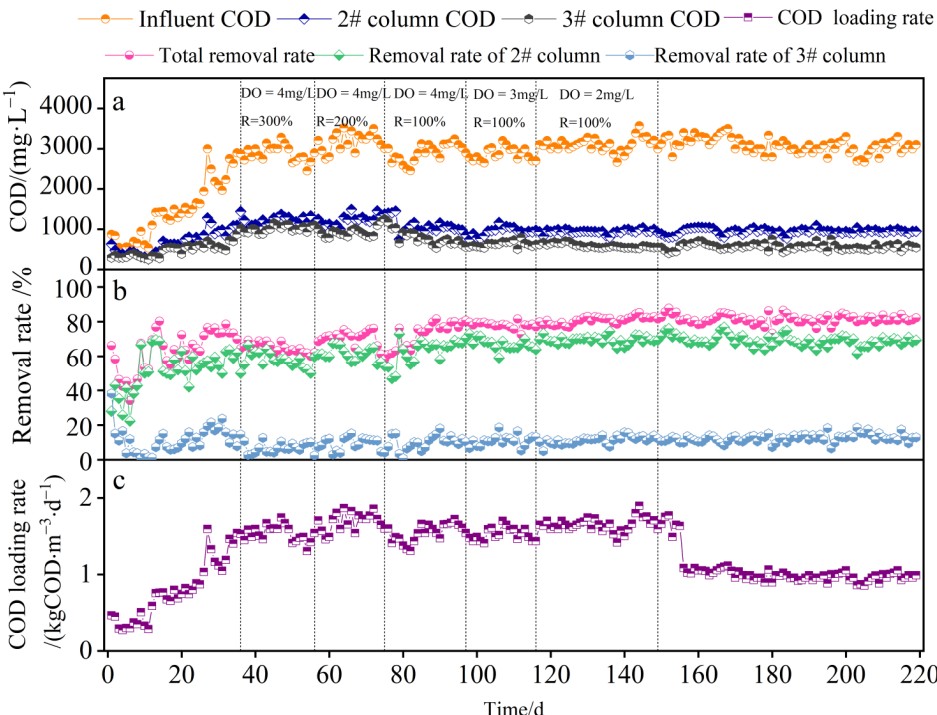

**Figure 3.** The COD removal in the simultaneous nitrification and denitrification unit in the debugging process: (**a**) influent COD in each reactor; (**b**) COD removal rate; (**c**) COD loading tate.

Day 36 to day 149 was the exploration stage of the operating conditions. During this stage, appropriate operating parameters were obtained by monitoring the COD removal in response to the changes in the influent loading and operating conditions. From the 36th to the 97th day, the total COD removal rate of the system gradually increased with the decrease in the reflux ratio (Figure 3). During the reflux ratio of 300%, 200%, and 100%, the total COD removal rates were observed to be 66.06%, 70.29%, and 73.19%, respectively. The corresponding effluent COD concentrations were 1011.61 mg/L, 938.02 mg/L, and 767.95 mg/L, respectively. Column-wise, the COD removal rates in Column 2# were 57.61%, 61.02%, and 63.07%, while the COD removal rates in Column 3# were 7.18%, 9.27%, and 12.32% for the reflux ratio of 300%, 200%, and 100%, respectively. This may be attributed to the fact that the decrease in the reflux ratio resulted in the prolonged hydraulic retention time of the wastewater in the system. Furthermore, the shock of the pollutant hydraulic loading on the system was reduced, which was beneficial for the microbial degradation of organic substances in the wastewater. In addition, the DO content in the wastewater entering Column 2# also decreased gradually with the decrease in reflux ratio [35]. This was conducive to the better denitrification process in Column 2#. This led to the increased demand of the organic carbon source for the denitrifying bacteria in Column 2#, thereby reducing the effluent COD concentration.

From the 75th to the 149th day, the decrease in the DO concentration in Column 3# resulted in the gradual increase in the total COD removal rate of the system (Figure 3). At the DO concentration of 4 mg/L, 3 mg/L, and 2 mg/L in Column 3#, the total COD removal rates of the system were 73.19%, 78.08%, and 80.22%, respectively, whereas the effluent COD concentrations were 767.95 mg/L, 628.4 mg/L, and 612.25 mg/L, respectively. The corresponding column-wise COD removal rates were 63.07%, 67.01%, and 69.47%, respectively (Column 2#), and 10.12%, 11.42%, and 12.32%, respectively (Column 3#). The metabolic activity of the microorganisms in Column 2# was inhibited by the increase in the DO concentration in the recirculation water of Column 3#. As a result, the denitrification process weakened, and the consumption of organic matter was reduced. Accordingly, the COD removal rate was also reduced.

Day 149 to day 219 was the connection stage. During this stage, the influent inflow was maintained, and the influent concentration was increased to be the same as the effluent concentration in the anaerobic reactor. At the reflux ratio of 100% and DO concentration of 2 mg/L, the SND units showed good COD removal for the alcohol precipitation wastewater of Chinese patent medicine treated by anaerobic digestion.

### 3.3. Pollutant Removal during the Coupled Process

3.3.1. Removal of Conventional Pollutants

After completing the debugging of the anaerobic digestion and SND units, a stable operation of the coupled process was conducted. The changes in the concentrations of conventional pollutants in the influent and effluent during the overall operation are shown in Figure 4a–c.

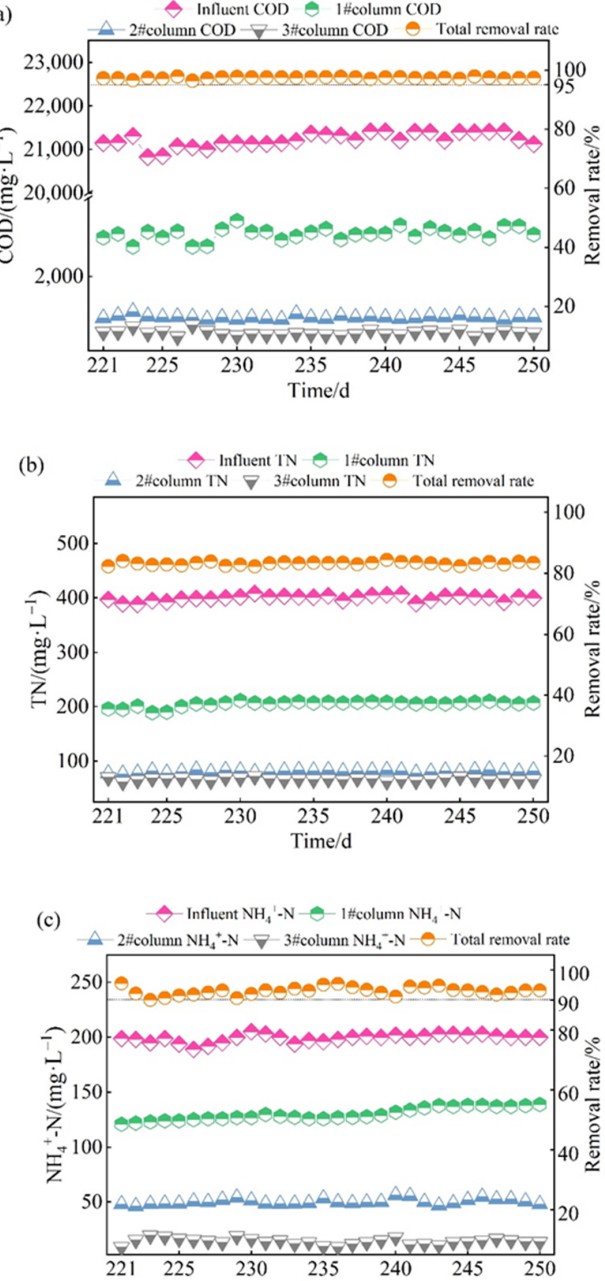

**Figure 4.** The removal rate during the coupled process: (**a**) COD; (**b**)TN; (**c**) NH$_4^+$-N.

During the overall operation of the process, the influent COD concentration was around 21,000 mg/L, the effluent COD concentration was generally less than 550 mg/L, and the COD removal rate was maintained at around 97% (Figure 4a). The influent TN concentration was approximately 400 mg/L, the effluent TN concentration was around 60 mg/L, and the TN removal rate was maintained at around 85% (Figure 4b). The concentrations of $NH_4^+$-N, nitrate ($NO_3$-N), nitrite ($NO_2$-N), and organic nitrogen in TN that had not been removed were detected to be 9 mg/L, 10 mg/L, 3 mg/L, and 38 mg/L, respectively. In addition, the influent $NH_4^+$-N concentration, effluent $NH_4^+$-N concentration, and $NH_4^+$-N removal rates were observed to be around 200 mg/L, 9 mg/L, and 96%, respectively (Figure 4c).

### 3.3.2. Removal of Lignin

The biodegradation of lignin in wastewater mainly relies on oxidoreductase enzymes such as peroxidase, manganese-dependent peroxidase, and laccase, produced by fungi and bacteria [36,37]. During the overall operation of the process, the changes observed in the lignin content of the influent and effluent are shown in Figure 5.

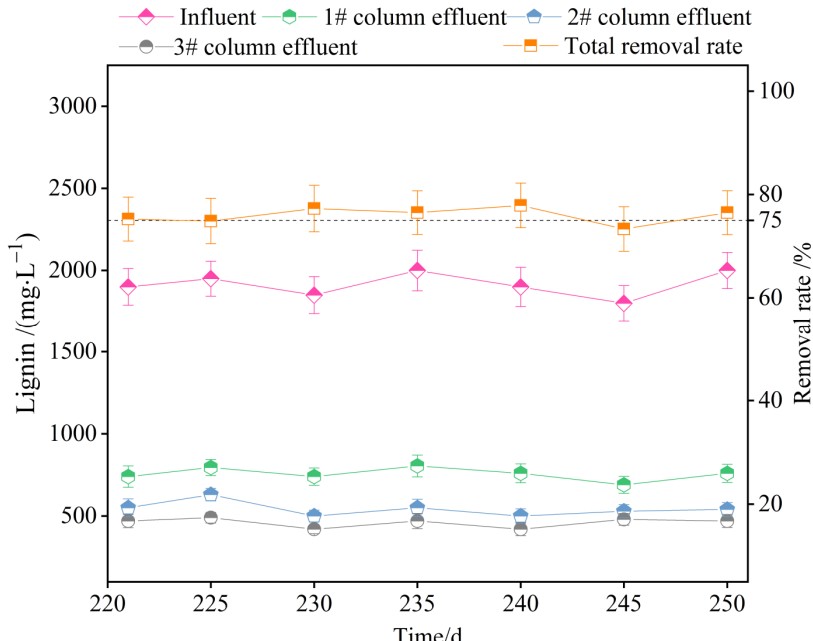

**Figure 5.** The lignin removal efficiency during the overall coupled process.

As shown in Figure 5, during the coupled process, the influent lignin was ~1800 mg/L, the effluent lignin in Column 3# was ~450 mg/L, and the total lignin removal rate was around 75%. The lignin removal rate in the anaerobic digestion unit was approximately 60%, which was higher than the removal rate in the SND unit. Higher lignin removal during anaerobic digestion can be attributed to three main reasons. First, lignin is removed through biodegradation, which is related to peroxidase, manganese-dependent peroxidase, and laccase produced by microorganisms such as *Pseudomonas*, *Nocardia*, *Achromobacter*, *Sphingobium*, and *Rhodococcus* [38,39]. The second reason for the lignin removal is that lignin can be carried by the gas bubbles such as biogas (produced in the anaerobic reactor) and floated up to the water surface, thereby being separated from the water [40]. The third reason is that lignin may have been dissolved into a colloidal form by ethanol in the alcohol precipitation wastewater, and then removed by flocculation and precipitation after the microbial action.

### 3.3.3. Acute-Toxicity Removal Effect

The wastewater used was the alcohol precipitation wastewater collected from the firm producing the Chinese patent medicine, using Sanchaku and Banlangen as raw materials, which had strong biological toxicity [41]. Hence, the acute toxicity of the influent and effluent in each unit was examined to investigate the tolerance and removal efficiency of the coupled process for toxicity caused by the wastewater (Figure 6).

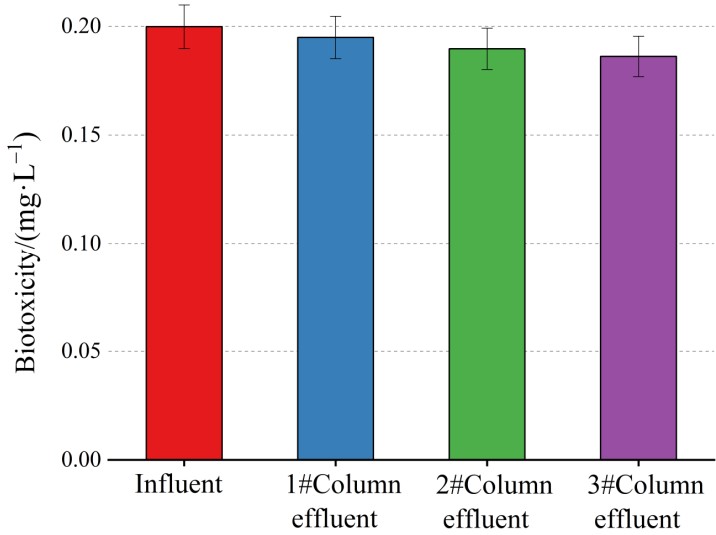

**Figure 6.** The acute-toxicity removal efficiency of the coupled process.

As shown in Figure 6, the acute toxicity of the influent during the overall operation of the process was 0.200 mg/L (based on the $HgCl_2$ toxic equivalent). After treatment, the acute toxicity of the effluent was 0.186 mg/L. The acute-toxicity removal rate was 7%, and the acute-toxicity intensity of Column 3# effluent was highly toxic (level V) [41]. Compared to a previous study [40], the process used in this study showed a certain removal effect and tolerance for the acute toxicity of the alcohol precipitation wastewater generated from the production of the Chinese patent medicines.

### 3.4. Changes in the Main Components of Wastewater

#### 3.4.1. 3D-EEM Analysis

During the overall operation of the coupled process, the influent (diluted 20 times) and the effluent of each unit were analyzed by 3D-EEM. The results are shown in Figure 7.

As shown in Figure 7a, corresponding to the influent, the 3D fluorescence Peak A appeared at $\lambda_{Ex}/\lambda_{Em}$ = 440~460 nm/500~540 nm, which was in the area V of the spectrogram and belonged to humic acid substances. The 3D fluorescence peaks in this area are generated by aromatic substances with stable properties and large molecular weight, which have obvious aromaticity and color [42]. This indicates that the DOM in the alcohol precipitation wastewater used in the study was mainly composed of humic acid substances. Most of these substances have aromatic ring structures and conjugated double bonds, which can inhibit microorganisms [43]. As shown in Figure 7b–d, the 3D fluorescence peaks (Peak B, Peak C, and Peak D) for effluents in Column 1#, 2# and 3# appeared at $\lambda_{Ex}/\lambda_{Em}$ = 420~460 nm/500~540 nm, $\lambda_{Ex}/\lambda_{Em}$ = 410~450 nm/480~520 nm, and $\lambda_{Ex}/\lambda_{Em}$ = 410~450 nm/480~520 nm, respectively. All the three peaks were in e region V of the spectrogram, which belonged to humic acid substances. This demonstrated that, after wastewater treatment by anaerobic digestion coupled with the SND process, the residual DOM still consisted of mainly humic acid substances.

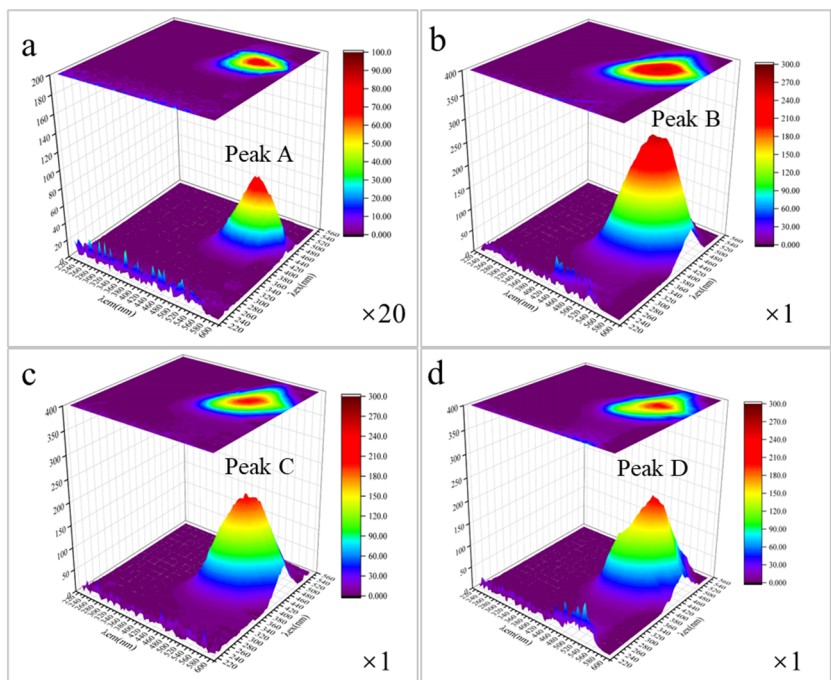

**Figure 7.** Excitation emission matrix spectra of (**a**) influent; (**b**) 1# column effluent; (**c**) 2# column effluent; (**d**) 3# column effluent.

Fluorescence intensity can characterize the changes in the content of humic acid substances in wastewater. The standard volume of fluorescence area integration calculated by the fluorescence area integration method can quantitatively characterize the DOM in wastewater [25]. Table 3 shows the fluorescence intensity and the standard volume of fluorescence area integration for the influent and effluent from three columns during the overall operation process. After the wastewater was treated, the removal rate of the fluorescence intensity was 86.47%, among which the removal rates in Column 1#, Column 2#, and Column 3# for influent fluorescence intensity were 82.72%, 3.34%, and 0.41%, respectively. This indicated that humic acid substances in wastewater were mainly removed in the anaerobic digestion unit. After the wastewater was treated by the system, the standard volume of the fluorescence area integration decreased by 77.59%, among which the standard volume of the fluorescence area integration for Column 1#, 2#, and 3# declined by 72.5%, 5.00%, and 0.09%, respectively. Combined with the blue-shift phenomenon (i.e., the fluorescence peak moved in the short-wavelength direction) in each unit, the results demonstrated that the recalcitrant substances were degraded.

**Table 3.** Fluorescence intensity and standard volume of fluorescence area integration.

| Items | Unit | Influent | 1# Column Effluent | 2# Column Effluent | 3# Column Effluent | Total Removal Rate |
|---|---|---|---|---|---|---|
| Fluorescence intensity | a.u | 1355.94 | 234.35 | 189.03 | 183.50 | 86.47% |
| Standard volume of fluorescence area integration | au·nm$^2$ | $1.16 \times 10^7$ | $3.19 \times 10^6$ | $2.61 \times 10^6$ | $2.60 \times 10^6$ | 77.59% |

### 3.4.2. FTIR Analysis

During the overall operation of the process, the detection results of FTIR for the influent and effluent of each unit are shown in Figure 8.

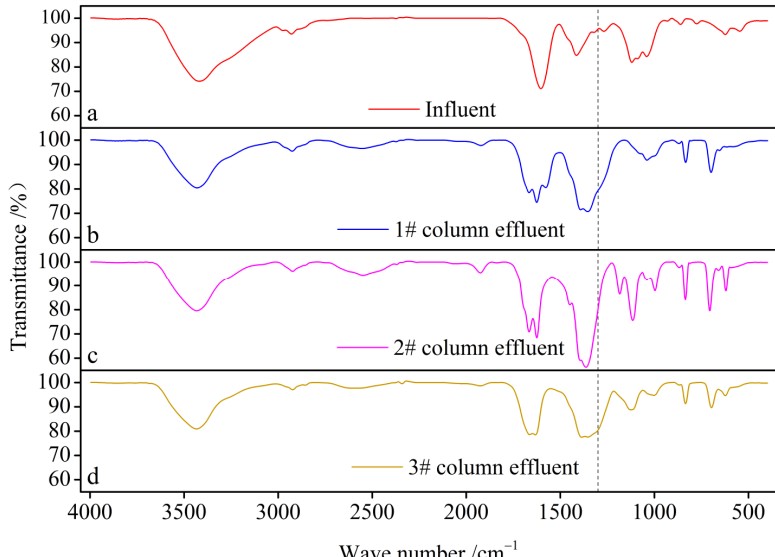

**Figure 8.** Fourier-transform infrared spectra of (**a**) influent; (**b**) 1# column effluent; (**c**) 2# column effluent; (**d**) 3# column effluent.

Figure 8a reveals the FTIR spectrum for the influent. In the functional group region, the peak at 3420.22 cm$^{-1}$ corresponded to the N-H and C-H stretching vibration bands, while the peak around 2929.27 cm$^{-1}$ denoted the C-H stretching vibration band. Peaks at 1602.87 cm$^{-1}$ and 1413.81 cm$^{-1}$ corresponded to the stretching vibration bands of aromatic compounds and the protein peptide bond, respectively [44]. In the fingerprint region, 1119.58 cm$^{-1}$ was the C-O-C in-plane bending vibration band of fats, which the characteristic peak of lignin [45]. On the other hand, the peak at 1040.84 cm$^{-1}$ represented the C-O-C in-plane bending vibration band of carbohydrates, which is the characteristic peak of xylan [45]. The peaks at 860.64 cm$^{-1}$ and 774.32 cm$^{-1}$ were the benzene ring substitution stretching vibration band and the O-H out-of-plane bending vibration band, respectively, which indicated the presence of phenolic compounds in the wastewater. Similarly, 622.59 cm$^{-1}$ and 546.57 cm$^{-1}$ were the C-Cl or C-Br stretching vibration band, which indicated the presence of halogenated hydrocarbons in the wastewater. Comprehensive analysis revealed that the structure of substances in the alcohol precipitation wastewater used in this study included benzene rings, saturated carbon bonds, unsaturated carbon bonds, hydroxyl groups, and ether bonds. Furthermore, the wastewater contained substances including alkanes, olefins, phenols, alcohols, halogenated hydrocarbons, amines, and aromatic compounds.

As shown in Figure 8b–d, for the effluents of three columns the stretching vibration bands at 3420 cm$^{-1}$ decreased as compared to the influent, indicating that the organic matter and nitrogen-containing substances in the wastewater were effectively removed after treatment. For the effluent of each unit, the stretching vibration bands of the carboxyl functional groups appeared near 2554.49 cm$^{-1}$ and 1924.85 cm$^{-1}$, which showed that the effluent contained small molecular carboxylic acid substances. This suggested that the macromolecular organic matter was converted into small molecular substances after biological metabolism. The stretching vibration band around 1602.87 cm$^{-1}$ for the effluent in each unit appeared with a split peak, indicating that the structure of aromatic compounds in wastewater changed after treatment, which can be related to the acute-toxicity removal from the wastewater. However, the characteristic peaks of phenolic compounds and halogenated hydrocarbon compounds in the fingerprint region did not change much. These phenomena were in agreement with the low acute-toxicity removal rate of the wastewater. The stretching vibration band appeared at 1184.00 cm$^{-1}$ for the effluent of Column 2#. This band was the stretching vibration band of ester compounds generated by the esterification reaction between the carboxylic acid and the incompletely utilized

alcohol during anaerobic digestion in Column 1#. The characteristic peaks of lignin and carbohydrates were reduced in the FTIR spectra for the effluent of each unit. The reduction in the characteristic peak of lignin was most obvious in Column 1# (i.e., the anaerobic digestion unit), which contributed a lot in the degradation of lignin. There were almost no characteristic peaks of carbohydrates in the spectra for the effluent of Column 3#. This indicated that microorganisms have a higher carbohydrate utilization rate in the absence of external carbon sources.

### 3.4.3. UV–Vis Absorption Spectroscopy Analysis

During the overall operation of the process, the influent and effluent of each unit were diluted 10 times and then analyzed using UV–Vis absorption spectroscopy. Figure 9 shows the results of the UV–Vis spectroscopy.

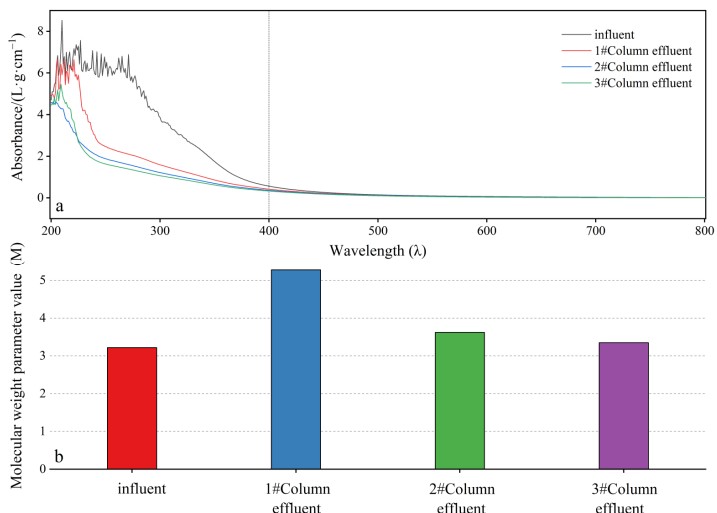

**Figure 9.** (**a**) Ultraviolet–visible absorption spectra and (**b**) M values for influent and effluents of three columns.

As shown in Figure 9a, the influent had a strong absorbance in the ultraviolet region, but almost zero absorbance in the visible region. This indicated that the organic matter in the alcohol precipitation wastewater used in the study was mainly unsaturated organic matter and aromatic compounds. Peaks at 206 nm, 214 nm, and 221 nm were the absorption peaks of aromatic rings, while the peaks at 210 nm and 217 nm were the absorption peaks of phenolic hydroxyl groups and chromophoric groups, respectively [46]. Absorption peaks at 223 nm and 227 nm corresponded to the conjugated polyenes and -C=C-C=O-, respectively [46]. Meanwhile, it can also be seen that the degradation of substances in the ultraviolet light region (200 nm~400 nm) of the effluent from each unit was remarkable.

In the UV–Vis absorption spectrum, the molecular weight parameter M is the ratio of absorbance at 250 nm to 365 nm. The value of M is inversely proportional to the molecular weight. The smaller the value is, the higher the proportion of macromolecular organic compounds would be [47]. As shown in Figure 9b, the M values for the influent, Column 1# effluent, Column 2# effluent, and Column 3# effluent were 3.22, 5.28, 3.62, and 3.35, respectively. These values indicated that anaerobic digestion coupled with the SND process had an obvious effect on the degradation of macromolecular organic matter in the wastewater.

## 4. Conclusions

A pilot study was carried out for the treatment of alcohol precipitation wastewater, by adopting anaerobic digestion coupled with the SND process. The study investigated the efficiency of this treatment process and explored the underlying mechanism of pollutant degradation. The main conclusions are as follows.

(1) After debugging for 220 days, the anaerobic digestion unit was operated at the influent volume loading of 10 kg COD/(m$^3$·d), and the SND units operated at the influent volume loading of 1.0 kg COD/(m$^3$·d) and 0.07 kg TN/(m$^3$·d). The overall treatment efficiency of the coupled process was stable. The COD removal rate, TN removal rate, and NH$_4^+$-N removal rate were 97%, 85%, and 96%, respectively. The lignin removal rate was 75%, while the acute-toxicity removal rate was 7%.

(2) The results of 3D-EEM, FTIR, and UV–Vis absorption spectroscopy showed that the wastewater contained alkanes, alkenes, phenols, alcohols, amines, halogenated hydrocarbons, unsaturated organic matter, aromatic compounds, and humic acid substances. Moreover, the DOM in the wastewater mainly consisted of humic acid substances.

(3) After the wastewater was treated by each unit, the 3D fluorescence peak intensity decreased by 86%, and the standard volume of the fluorescence area integration declined by 78%. These observations indicated that the humic acid substances were degraded in the process. The stretching vibration band of aromatic compounds in the effluent of each unit appeared with split peaks, demonstrating that the biological toxicity and aromaticity of the wastewater tended to decrease. The molecular weight parameter value in the ultraviolet region increased, indicating that the removal of macromolecular organic matter in the wastewater was quite significant.

**Author Contributions:** X.J.: Writing—original draft; data curation. H.F.: investigation; carrying out experiments; data curation. J.L.: supervision; writing—review and editing. J.D.: project administration; resources. Y.S.: methodology; editing. W.M.: methodology; editing. All authors have read and agreed to the published version of the manuscript.

**Funding:** This research was funded by National Natural Science Foundation of China (No. 41902266), and Science and Technology Project of Henan province (222102320152, 212102310520).

**Data Availability Statement:** The data presented in this study are available on request from the corresponding author.

**Acknowledgments:** This work was supported by the National Natural Science Foundation of China (No. 41902266), and the Science and Technology Project of Henan province (222102320152, 212102310520). The authors would like to thank all the reviewers who participated in the review.

**Conflicts of Interest:** The authors declare no conflict of interest.

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
