# Peer review of "Performance Analysis of Anaerobic Digestion Coupled with Simultaneous Nitrification and Denitrification Process for Treating Alcohol Precipitation Wastewater of Chinese Patent Medicine"

_water, doi:10.3390/w15101939_

Round 1

Reviewer 1 Report

The manuscript is focused on the performance analysis of anaerobic digestion coupled with nitrification and denitrification process for treating alcohol wastewater.

The work is interesting, however, before the publication in the Water journal it should be improved. For this purpose, please, see the comments below:

1. The abstract is too long. Indeed, according to instructions of authors,  the abstract should be a total of about 200 words maximum. 

2. The introduction does not  provide sufficient background and does not include all relevant references:

- The aim of the work should be clearly presented.

- What is the novelty of this work? What is the difference between the work performed and others studies related to this subject previously published?

- The literature review should be performed better. Indeed, the Authors cited only 37 papers. Only -of them have been published in the last 3 years. This point should be improved significantly.

3. The Conclusion section is definitely too long. It should present only the most important results and perspectives.

4. Another weak point of the work is that the Authors did not compare the obtained results with those available in the literature.

5. The manuscript is written carelessly, for example, there is no space before the number of the cited work.

6. Minor editing of English language is required.

7. The quality of the Figure 3 should be improved.

To sum up, the work is interesting, however, please improve it according to presented sugestions. I suggest major revision.

Minor editing of English language is required.

Reviewer 2 Report

In this manuscript the authors have reported for treating alcohol precipitation wastewater of Chinese medicine. The experimental work is comprehensive and presented in detail. The manuscript is written well and there are some minor changes which are highlighted in the PDF file attached. The authors should address the recommended changes

The English is fine, I have not seen any incorrect spelling or grammatic mistake in the manuscript. 

Reviewer 3 Report

I’ve just finished a review of the paper water-2344953, titled: “Performance analysis of anaerobic digestion coupled with simultaneous nitrification and denitrification process for treating alcohol precipitation wastewater of Chinese patent medicine” and written by the authors: Xiaofeng Jia, Hanxin Fan, Jiawei Liang, Jihua Dai, Yu Sun, Wenning Mai.

In this paper, the biological treatment method of anaerobic digestion coupled with simultaneous nitrification and denitrification was adopted to investigate its efficiency for treating alcohol precipitation wastewater and to explore the mechanism of pollutant degradation in this process. The results showed that after 220 days of debugging, the anaerobic digestion coupled with simultaneous nitrification and denitrification process operated stably. The influent COD, total nitrogen (TN), ammonium (NH4+-N), and lignin concentrations were 21000 mg/L, 400 mg/L, 200 mg/L, and 1800 mg/L, respectively. The removal rates of COD, TN, NH4+-N, and lignin were 97%, 85%, 96%, and 75%, respectively. Furthermore, the acute toxicity of the influent was 0.2 mg/L, and the acute toxicity reduction amount was 0.014 mg/L.

In general, the paper is interesting for readers, and the topic of the paper is actual and in agreement with the topic of the journal Water. The results are quite well described, the English is correct. From that point, I do not have any significant remark and thus I suggest its acceptance for publication. However, there are some things that should be corrected in the paper before publication, and they are as follows:

1.      In Table 1 please check the values for the COD. Is 300.000 mg/L correct?

2.      In lines 160-163: Please add a description for the FTIR and UV-VIS methods.

3.      In the paper authors did not mention what is the capacity of the applied method for potential practical application. So how many liters of water which is with a defined percentage of the contamination, can be purified with one cycle of purification? How many cycles one biological treatment agens can be used? How much biological treatment agens is required for one cycle of purification? Also, how much is expensive purification in this way? Industrial scale calculation (observed in practice or theoretical) is required to be inserted in the paper.

Reviewer 4 Report

Find the specific comment to the authors for more details.

1.     Page 3, Line 105-106, “The size of the boĴom reactionzone was Φ100 mmÍ600 mm, and 105 the size of the top sedimentation zone wasΦ200 mmÍ200 mm”. The statement has been revised accordingly.

2.     What are the substrate and inoculum loading rates (VS basis)? How many replicates for each treatment?

3.     Page 8, line 175-176 “Figure 2 shows the changes in influent and effluent COD, COD removal rate, and COD loading rate in each stage. Are soluble COD  or total COD present in these results? COD degradation kinetic data need to be provided. 

4.     What is the total solids content in the current study? 

5.     Page 6, Line 199-200 “To avoid reactor collapse due to continuous acidification, sodium hydroxide was added into the reactor, reducing influent in flow and influent concentration, and increasing effluent reflux production…Need to provide NaOH concentration and time interval.

6.     Page 7, Line 234-235 “the influent 234 COD was gradually increased from 500 mg/L to 3000 mg/L”. Please check.  

7.     Page 8, Line 261-265, “In addition, the DO content in the wastewater entering column 2# also decreased gradually with the decrease in reflux ratio. This was conducive for the beĴer denitrification process in column2#. This led to the increased demand of organic carbon source for denitrifying bacteria in column 2#, thereby reducing the effluent COD concentration. Please cite the reference for these statements. 

8.     Page 10, Line 309-311, “laccase produced by microorganisms such as Pseudomonas, Nocardia, Achromobacter, Sphingobium, and Rhodococcus[38,39]”. Microorganism name in italic formate…

Reviewer 5 Report

Interesting work that describes the treatment of Chinese patent medicine wastewater with a process consisting of anaerobic digestion and integrated denitrification and nitrification. The study is divided in two stages, the independent start up of the anaerobic and the coupled SND reactor (this was called the debugging) and the main porpoise was to determine de operating conditions before putting them in series.

The problem is that it is a complex process which is not fully described be just plotting the exit concentrations of COD and ammonia and total N in each reactor and in order to contribute to science and be useful for the readers more operating details are needed based in mass balances (Flow rates, dilution factors, recycle ratios, CH4, N2 produced together with the COD, and ammonia and nitrite, nitrate.

Please revise these specific suggestions:

Title. It is not simultaneous nitrification and denitrification because they are occurring in two separate reactors. Simultaneous DN suggests that they are both taking place in a single reactor (https://doi.org/10.1016/j.biortech.2015.11.088). In the text “the Chinese patent medicine” producing “alcohol precipitation” wastewater is well explained and even used indistinctly. There is no need to put them together here.

Abstract.

The full wastewater characterization and the dilution used in the study should be commented here. The purpose and results of the debugging stage should also be mentioned here. The removal efficiencies (incorrectly called rates) should be given for each reactor. For example, how much of the COD was removed in the anaerobic reactor and how much in the denitrifying reactor, how much N2 is produced here, how much of the NH4 is converted to NO3, how much it is recycled. Acute toxicity of a wastewater is difficult to be measured in terms of a single component. How is this concentration interpreted. It is not described in material and methods.

Keywords:

As they are used for searching and visibility purposes, should not repeat the words used in the title.

Introduction

Lines 73 to 86. State the differences between the work of Chen et al. 2008, Zhang et al. 2008 with this work because in this work two reactors were used.

Line 91. What is meant by long term debugging and what was the objective.

2. Material and methods

Experimental device

Line 115 State the aeration rate or criteria for airation

For the benefit of the reader, shortly explain what is shown in figure1. For example: Figure 1 Schematic diagram of the anaerobic digestion and integrated nitrification (#2)denitrification (#2).

What are the egg shaped figures beside columns 1 and 2?

222 Seed sludge.

Line 136 What was the MLSS in the anaerobic reactor?

2.3 line 14. Is debugging the right word to “setting the optimum operating conditions” or “start up”.

Explain the method used to measure acute toxicity.

Results and discussion

Figure 2. What determines the length of each stage. Are they relevant to mention? Figure 2b is usually plotted as the removal efficiency (not rate) as a function of COD loading rate. Gives more information.

Line 195 The influent concetration is not constant in stage II as shown in figure 2a.

Figure 3. The COD removal is performed in colun#2 because NO3 is recycled to oxydize the remaining organic matter from the anaerobic reactor. NO3 is reduced to N2. Column#3 oxydizes ammonia to NO3 under low COD. That is the reason to ask for ithe individual reactor performance.

3.3 1 and 3.3.2 Removal of conventional pollutants and lignin.

Figures 4 to 7 show no variation in this integrated operation, so it would be more illustrative to state these values in a diagram that will show the performance of each reactor.

3.3.3 Accute toxicity. Explain the methods in material and methods. Results are not clear. Table 1 says 2 to 3 mg/L based on HgCl2 while figure 8 says 0.2. What is the toxic material in this effluent.

A table summarizing figures 9, 10 & 11 as to which substances are removed in which reactor will be very conclusive and illustrative and will match with the proposed diagram to integrate figures 4 to 7.

Very good English.

Round 2

Reviewer 1 Report

The manuscript hqs been improved according to my comments, hence, I recommend it for publication.

Author Response

Thank you very much for your review and affirmation.

Reviewer 3 Report

The paper should be accepted in its present form.

Best regards

Author Response

(The authors gave the same response as above.)
